# COVID-19 Pandemic and Remote Consultations in Children: A Bibliometric Analysis

**DOI:** 10.3390/ijerph19169787

**Published:** 2022-08-09

**Authors:** Nicole Camoni, Silvia Cirio, Claudia Salerno, Araxi Balian, Giulia Bruni, Valeria D’Avola, Maria Grazia Cagetti

**Affiliations:** 1Department of Restorative, Preventive and Paediatric Dentistry, School of Dental Medicine, University of Bern, 3010 Bern, Switzerland; 2Department of Biomedical, Surgical and Dental Sciences, University of Milan, 20122 Milan, Italy; 3Dental and Stomatology Unit, Cittadella Hospital, AULSS 6 Euganea, 35013 Cittadella, Italy

**Keywords:** telemedicine, bibliometric analysis, non-communicable diseases, pediatrics

## Abstract

Telemedicine is becoming a standard method of consultation, and the COVID-19 pandemic has increased its need. Telemedicine is suitable for non-communicable diseases (NCDs) in the pediatric population, as these are chronic conditions that affect many children worldwide. The aim of this study was to analyze the bibliometric parameters of publications on the use of telemedicine for the most common NCDs in children before and after the COVID-19 pandemic. Following the electronic search, 585 records were selected. “Metabolic diseases” was the most frequent topic before and after the pandemic, accounting for 34.76% in 2017–2019 and 33.97% in 2020–2022. The average IF of the journals from which records were retrieved was 5.46 ± 4.62 before and 4.58 ± 2.82 after the pandemic, with no significant variation. The number of citations per reference averaged 14.71 ± 17.16 in 2017–2019 (95% CI: 12.07; 17.36) and 5.54 ± 13.71 in 2020–2022 (95% CI: 4.23; 6.86). Asthma, metabolic diseases, and neurodevelopmental disorders were the most explored topics. A relevant finding concerns the increasing number of observational studies after the pandemic, with a reduction of the interventional studies. The latter type of study should be recommended as it can increase the evaluation of new strategies for the management of NCDs.

## 1. Introduction

Health technologies for remote consultation and treatment are improving outcomes for patients and clinicians, especially in recent years, when high-quality telehealth and telemedicine techniques have become more feasible and user friendly [1,2,3].

Telehealth and telemedicine are now often used as synonyms, although they were conceived as separate terms. Telehealth refers to the use of telecommunications to promote healthcare at a distance (including pre-recorded conferences and courses for health professionals), while telemedicine is a term with a narrower meaning used only for real-time communication between practitioners and patients [4]. Telemedicine includes “hospital” consultations for emergencies, adopted in situations where the care unit requires remote consultation for hyper-specialized cases or to avoid unnecessary transfers; “ambulatory” consultations are adopted for less urgent cases [5,6].

Although telehealth services were arising in the pre-pandemic period, the 2019 outbreak of COVID-19 deeply modified healthcare worldwide [7]. The lockdowns, with the closure of some care services and the need for non-emergency check-ups, stimulated the implementation of telemedicine tools [8]. Moreover, both patients and physicians were motivated to minimize in-person contact to prevent the spread of the coronavirus, and telemedicine again offered a practical solution for providing consultations [9].

Many therapies and diagnoses can be performed not in person and, for chronic patients, the possibility of maintaining regular follow ups from home is practical and inexpensive [10]. In the course of a pandemic, telemedicine helps to avoid unnecessary contact between chronically ill children and physicians, as consultations with pediatric patients require the presence of a parent/caregiver and this means a potential prolonged contact of at least three subjects for consultation [11,12]. To prevent contact and to minimize the relative risk of contagion among the team of practitioners, safety procedures have been proposed, as, with new waves, it may be possible to avoid massive contagions among the clinical staff, particularly physicians [13]. In addition, telemedicine is well positioned during this time to reduce the potential spread of the disease and prevent overloading of the healthcare system through at-home COVID-19 screening, diagnosis, and monitoring. New technologies for symptom monitoring can control future COVID-19 infection peaks and optimize patient outcomes [8].

The use of telecare/telemedicine is particularly suitable for the management of non-communicable diseases (NCDs) in the pediatric population. Teleconsultations for chronic patients already existed before the COVID-19 pandemic, especially in rural communities [14,15]. With regard to NCDs, children represent a major global burden, with a wide range of conditions from diabetes, cancer, asthma, congenital defects, and behavioral disorders to dental caries [16]. In the adult population, the WHO has stated that 40 million deaths per year are due to these diseases, which could be at least partially reduced if effective and continuous counseling was provided at an early age [17,18,19]. Many NCDs begin in childhood or early adolescence, when the body is particularly vulnerable to the same risk factors responsible for chronic diseases in adults, such as exposure to tobacco or alcohol; diets high in fat, salt, and sugar; and physical inactivity [19]. For this reason, telehealth for NCDs in children needs to be supported and implemented by healthcare professionals. The scientific literature reflects these social and clinical facts with increasing studies on technological tools for patients, caregivers, and physicians [20,21]. Nevertheless, it is not always possible to find out what the general trend is for a specific medical topic or whether important aspects of this same topic are missing in the literature, causing insufficient dissemination of information [22]. Knowing the trend of articles and the performance of scientific journals on a specific topic is useful for summarizing current bibliometric findings and directing future lines of research [23,24]. These objectives can be achieved through a scoping review. A scoping review is a relatively new research methodology that is particularly effective for examining the extent, range, and nature of research activity. This type of review does not describe the findings of the research in detail but is a useful way of mapping fields of study where it is difficult to visualize the range of material that might be available [25].

Therefore, the aim of this study is to analyze the bibliometric parameters of articles that evaluated the use of telemedicine for the most prevalent NCDs in children before and after the COVID-19 pandemic. The authors decided to compare data from the second half of 2017 to 2019 with data from 2020 to the first six months of 2022 to outline the main trends in the literature during these periods.

## 2. Materials and Methods

### 2.1. Study Design

The PRISMA extension for scoping reviews (PRISMA-ScR) was followed (Appendix A) [26]. A two-step process was used to cover all aspects of the research objectives. Step one: Conduct a scoping review adopting the Arksey and O’Malley framework [25]. Step two: Conduct a bibliometric analysis of the scoping review results to analyze the knowledge domains and possible future research trends. The research question was formulated according to the following PICO: P (population): articles on children with common NCDs; I (intervention): number and type of articles published on the use of telemedicine after the COVID-19 pandemic; C (comparison): number and type of articles published before the COVID-19 pandemic; O (outcome): quantitative bibliometric data.

#### 2.1.1. Inclusion Criteria

The following inclusion criteria were adopted:Publication date from 1 July 2017 to 30 June 2022;Abstracts, conference papers, and editorial written in English; papers written in any language with an abstract in English;Papers published in journals, books, or conference booklets;Studies including pediatric populations.

#### 2.1.2. Exclusion Criteria

The following exclusion criteria were adopted:Publication date not in the inclusion criteria range;Studies that only considered adult populations.

#### 2.1.3. Electronic Search

The search was restricted to the 50 most common NCDs in children in high-income countries, and the Global Health Data Exchange (GHDx) catalog was consulted for this purpose on 12 February 2022. Among the 50 most frequent NCDs category, 31 were excluded since they referred to broader disease entities (e.g., “other neoplasms”, “pain”) rather than specific diseases. In total, 19 NCDs were selected and considered for further electronic search. These are, from the most common to the least common: caries of deciduous teeth, tension-type headache, caries of permanent teeth, asthma, atopic dermatitis, migraine, acne vulgaris, G6PD deficiency, attention-deficit/hyperactivity disorder, premenstrual syndrome, anxiety disorders, conduct disorder, autism spectrum disorders, major depressive disorder, pyoderma, psoriasis, contact dermatitis, epilepsy, and diabetes mellitus type I.

The electronic search was conducted on 3 databases: PubMed (National Library of Medicine), Web of Science (Clarivate Analytics), and Embase (Elsevier) by two independent authors (N.C. and A.B.). The research was conducted in February 2022 and updated on 1 August 2022. The following search strings were used:for Medline: (“non communicable disease-” [tiab] OR “NCDs” [tiab] OR “caries” [tiab] OR “decay” [tiab] OR “G6PD deficiency” [tiab] OR “favism” [tiab] OR “atopic dermatitis” [tiab] OR “acne” [tiab] OR “premenstrual syndrome” [tiab] OR “tension type headache” [tiab] OR “anxiety disorder-” [tiab] OR “adhd” [tiab] OR “migraine” [tiab] OR “asthma” [tiab] OR “conduct disorder-” [tiab] OR “autism” [tiab] OR “major depression” [tiab] OR “major depressive disorder*” [tiab] OR “pyoderma” [tiab] OR “psoriasis” [tiab] OR “contact dermatitis” [tiab] OR “epilepsy” [tiab] OR “diabetes” [tiab]) AND (“telemedicine” [mh] OR “remote consultation” [tiab] OR “telemedicine” [tiab] OR “tele health” [tiab] OR “tele-health” [tiab] OR “telehealth” [tiab] OR “remote patient-” [tiab] OR “teleclinics” [tiab]) NOT (“pregnant” [tiab] OR “pregnancy” [tiab]);for Embase: (‘non communicable diseases’:ab OR ‘ncds’:ab OR ‘caries’:ab OR ‘decay’:ab OR ‘g6pd deficiency’:ab OR ‘favism’:ab OR ‘atopic dermatitis’:ab OR ‘acne’:ab OR ‘conduct disorder*’:ab OR ‘contact dermatitis’:ab OR ‘pyoderma’:ab OR ‘psoriasis’:ab OR ‘premenstrual syndrome’:ab OR ‘tension type headache’:ab OR ‘anxiety disorders’:ab OR ‘adhd’:ab OR ‘autism’:ab OR ‘epilepsy’:ab OR ‘major depression’:ab OR ‘major depressive disorder’:ab OR ‘migraine’:ab OR ‘asthma’:ab OR ‘diabetes’:ab) AND (‘remote consultation’:ab OR ‘telemedicine’:ab OR ‘tele health’:ab OR ‘tele-health’:ab OR ‘telehealth’:ab OR ‘remote patient-’:ab) NOT (‘pregnant’:ab OR ‘pregnancy’:ab) AND ([newborn]/lim OR [infant]/lim OR [child]/lim OR [preschool]/lim OR [school]/lim OR [adolescent]/lim);for Web of Science: ((ts = (non communicable diseases OR dental caries OR glucose 6 phosphate dehydrogenase OR favism OR atopic dermatitis OR acne OR pyoderma OR premenstrual syndrome OR tension headache OR anxiety disorder OR adhd OR major depression OR migraine OR asthma OR diabetes OR conduct disorder OR psoriasis OR contact dermatitis OR autism OR epilepsy OR diabetes)) AND ALL = (telemedicine)) AND ALL = (child* OR pediatri*).

The Clarivate Analytics Journal Citation Report (JCR) (Clarivate Analytics) was consulted to retrieve bibliometric information of academic journals from which selected references were searched.

### 2.2. Data Analysis

All retrieved references were uploaded to a Microsoft Office Excel^®^ spreadsheet to check for duplicates and select studies for inclusion. After the exclusion of duplicates (767 studies), two independent authors (S.C. and V.D.) screened the records by title and abstract; in case of concern, a consensus was reached after consultation with a third author (M.G.C.). The agreement between the two selectors was assessed using Cohen’s kappa score.

The following data were collected for each record: title, authors, year, country, number of citations, journal title, journals’ JCR 2020 impact factor, journals’ JCR subject category.

Each reference was also classified according to the following criteria:Type of publication: article, conference paper, conference abstract, conference review, book chapter, or thesis.Main topic (i.e., which NCD). The authors labeled the NCDs according to PubMed’s medical subject headings in the following topics: asthma, brain disorders, mental disorders, dental caries, metabolic diseases, neurodevelopmental diseases or disorders, and skin diseases.Type of study: case report/series; guidelines/consensus paper; interventional studies; observational studies; review/meta-analysis; study protocol/pilot study.Clinical query: diagnosis, prevention, treatment, follow up, and diagnosis plus treatment/follow up.

### 2.3. Statistical and Visual Analysis

Descriptive analysis was performed with EasyMedStat^®^ (Levallois-Perret, Francia). The normality and heteroskedasticity of the continuous data were assessed with Shapiro–Wilk and Levene’s tests, respectively. The continuous outcomes were compared with ANOVA, Welch’s ANOVA, or Kruskal–Wallis tests according to the distribution of the data. Discrete outcomes were compared with chi-squared or Fisher’s exact test. Alpha risk was set to 5%, and two-tailed tests were used. Google Colab and the VOSviewer tool were used to visualize the bibliometric data.

## 3. Results

The electronic search identified 1459 records; 692 were selected after removing duplicates, and then 107 papers were excluded after evaluation of the title and abstract because they did not meet the inclusion criteria or were identified as search noise. Finally, 585 records were included (Appendix A): 463 journal articles, 1 book chapter, 18 editorials/commentary/letters, and 103 congress abstracts.

Figure 1 shows the PRISMA flow diagram of the selection and classification process by topic of the included records.

Cohen’s kappa value for agreement between the two reviewers was 0.67 at title and abstract screening (94.6% agreement) and 0.87 at second screening (94.37% agreement).

Total records increased over the years reviewed: 27 in the second half of 2017, 72 in 2018, 65 in 2019, 126 in 2020, 220 in 2021, and 75 in the first six months of 2022. A percentage increase of 205.55% was found considering the two complete years (2018 and 2021) at the extremes of the time period considered.

More than half of all records were produced in North America and about a quarter in Europe both before and after the pandemic (Table 1). However, the distribution changed significantly after the pandemic (*p* = 0.001) due to the increase in records produced in Asia (from 4.26% in 2017–19 to 10.93% in 2020–22) and the concomitant decrease in those from Oceania (from 10.37% in 2017–2019 to 2.61% in 2020–2022) (Table 1; Figure 2).

The average IF of the journals from which the records were retrieved was 5.46 ± 4.62 before and 4.58 ± 2.82 after the pandemic, with no significant variation. The number of citations per reference averaged 14.71 ± 17.16 in 2017–2019 (95% CI: 12.07; 17.36) and 5.54 ± 13.71 in 2020–2022 (95% CI: 4.23; 6.86) (Table 2).

Journal article was the most frequent type of publication, followed by conference abstract, both before and after pandemic, with 80.49% and 14.63% in 2017–2019 and 78.62% and 18.76% in 2020–2022, respectively (Table 1). The distribution of the type of publication did not change significantly after the pandemic (*p* = 0.183).

Metabolic diseases were the most frequent topic both before and after the pandemic, accounting for 34.76% in 2017–19 and 33.97% in 2020–22 of all reviewed records (Table 1). The distribution of topics covered by the papers changed significantly after the pandemic (*p* < 0.001), mainly involving a decrease in the frequencies of asthma, dental caries, and mental disorders (from 23.78%, 7.32%, and 10.98% to 18.29%, 3.09%, and 6.18%, respectively), while the presence of topics related to brain and neurodevelopmental disorders increased (from 5.03% and 15.24% to 10.45% and 25.65%, respectively) (Table 1).

Endocrinology and metabolism and Pediatrics were the most frequent JCR journal category before (10.95% and 11.68%, respectively) and after (19.23% for both) the pandemic (Appendix A), while Health care sciences and service and Respiratory system decreased from 10.95% and 7.30% to 6.87% and 1.65%, respectively.

Interventional study was the most frequent type of study before the pandemic, followed by observational study and review/metanalysis (Table 1; Figure 3). The distribution of study type changed significantly after the pandemic: interventional study decreased from 50.00% in 2017–2019 to 31.12% in 2020–2022, while observational study and review/metanalysis increased from 18.90% to 34.68% and from 20.12% to 25.65%, respectively.

The most frequent clinical query was “treatment” (44.51%) before the pandemic and “follow-up” (41.81%) after (Table 1). The distribution of clinical queries did not change significantly, although an increase of “follow-up” was observed after the pandemic (from 30.49% to 41.81%).

Regarding the frequency of co-occurrence of terms in the title and abstract (more than 200 times) of the selected records, as shown in Figure 4, the words “telemedicine” and “telehealth” increased their occurrence from 2020 onward, while “intervention” was particularly used in the pre-pandemic era. A further bibliometric visualization of the type of studies and main topics can be found in Appendix A.

## 4. Discussion

Scientific output on the use of telemedicine for the most frequent NDSs in children increased significantly after the COVID-19 pandemic, as revealed by the bibliometric analysis performed from 2017 to 2022.

The records included in this bibliometric study provide an overview of how telemedicine scientific records have developed after the COVID-19 pandemic. A significant number of papers were found prior to the COVID-19 pandemic, suggesting that telemedicine was already a growing reality before the pandemic. However, the significant increase in the number of papers published in the years 2020–2022 revealed that the COVID-19 pandemic increased interest in telemedicine: this is likely a consequence of the need to switch to remote consultations and research activities due to restrictions that reduced access to hospitals, research centers, and academic institutions [7,27].

The good-to-excellent IF of the journals from which the records were retrieved, with an average value of more than three points, confirms that telemedicine is a popular topic and is addressed by a high level of scientific production. This is confirmed by the average number of citations, ranging from good to excellent, achieved by the selected references, between 5 and 10, also in view of the short period of time since their publication.

The predominance of publications dealing with metabolic diseases as the main topic and which were published in journals ranked in the Endocrinology and metabolism and Pediatrics JCR category is probably a consequence of an earlier application of telemedicine to non-communicable diseases such as diabetes [28]. However, the significant increase in publications concerning brain and neurodevelopmental diseases reveals that new applications of telemedicine have been investigated since the COVID-19 pandemic, especially for those diseases that show a worsening of patient’s conditions as a consequence of the social estrangement and lack of access to face-to-face healthcare [29].

Furthermore, it can be also assumed that diseases with a strong literature base will increasingly make use of telemedicine. This growing opportunity will reduce costs and unnecessary transfers, supporting families who bear the triple physical, psychological, and economic burden of a child’s chronic illness [2].

However, the use of telemedicine needs to be implemented for the most neglected NCD, namely dental caries, which accounts for a small number of records despite its high prevalence worldwide [30]. Although the topic dental caries showed an increasing trend in this research, data comparison revealed that teleconsultations for oral health problems are still insufficient and far from being a common practice before and after the COVID-19 pandemic. Caries prevention through the dissemination of information on good oral hygiene practice and low-sugar diets can easily be implemented if governments set up community-based telehealth projects [31,32].

The observed reduction in interventional studies and increase in observational studies can be explained by the challenges the pandemic has imposed on researchers and scientists around the world: access to hospitals and laboratories has been banned for months, limiting the planning and conduct of new trials.

A key process for the use of telemedicine for NCDs has already been stated by the WHO, which in 2021 addressed the need to improve the large-scale adoption of telehealth [33]. Specifically, the WHO’s future goals are to identify how the challenges to the NCD response brought by the COVID-19 pandemic can be addressed through digital solutions, to disseminate good practices and innovations in the use of digital health for NCDs prevention and monitoring, and to determine how the WHO can support countries to implement digital health solutions to better address the pandemic. The present research contributes in part to the WHO’s aim, as it summarizes trends in the use of telemedicine for NCDs in children. Additionally, it can be expected that many of the congress abstracts will became actual papers in the near future, allowing for more-precise conclusions to be drawn on the topics considered.

This study has some limitations. Firstly, the data are heterogeneous because the topics of the included records are very different from each other, as are the telemedicine solutions adopted. Secondly, this study summarizes the overall use of telemedicine in children with NCDs without analyzing each topic in detail and without assessing the results obtained from each paper. However, as this is a bibliometric review, this limitation is inherent in the type of paper. Furthermore, the fact that COVID-19 is still a global problem for clinical settings will make it necessary to update the findings with future reviews.

## 5. Conclusions

This bibliometric review confirms the pushing effect of the COVID-19 pandemic on the telemedicine literature. Asthma, metabolic diseases (in particular diabetes mellitus), and neurodevelopmental disorders (in particular autism and attention-deficit/hyperactivity disorder) are the most studied conditions for remote consultations. A relevant finding concerns the increasing number of observational studies since the COVID-19 pandemic but the decrease in interventional studies. This result leads the authors to hope that more interventional studies will be conducted in the near future as they could offer new ways of diagnosis, prevention, treatment, and follow up of NCDs, especially in children. Furthermore, more studies on the most neglected pediatric NCDs, such as dental caries and brain disorders, are desirable.

## Figures and Tables

**Figure 1 ijerph-19-09787-f001:**
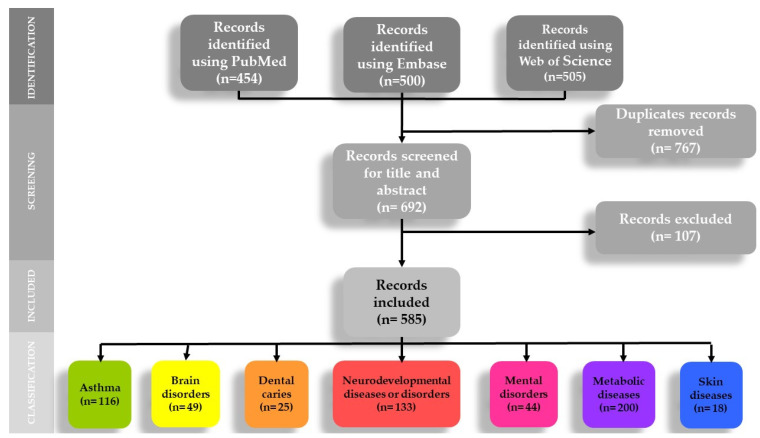
PRISMA flowchart of the selection process and classification process by topic of the included records.

**Figure 2 ijerph-19-09787-f002:**
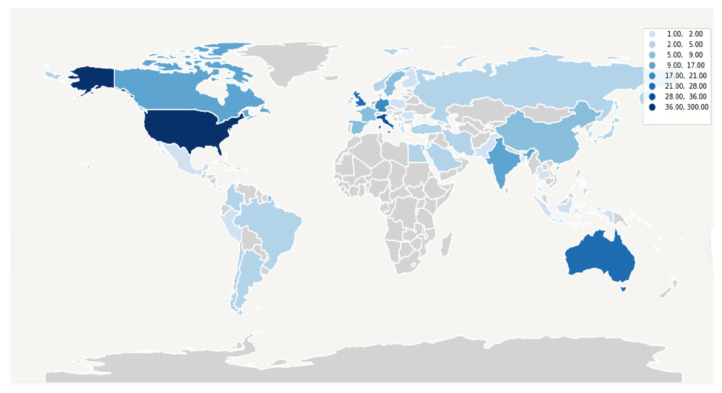
Prevalence of records published in different countries.

**Figure 3 ijerph-19-09787-f003:**
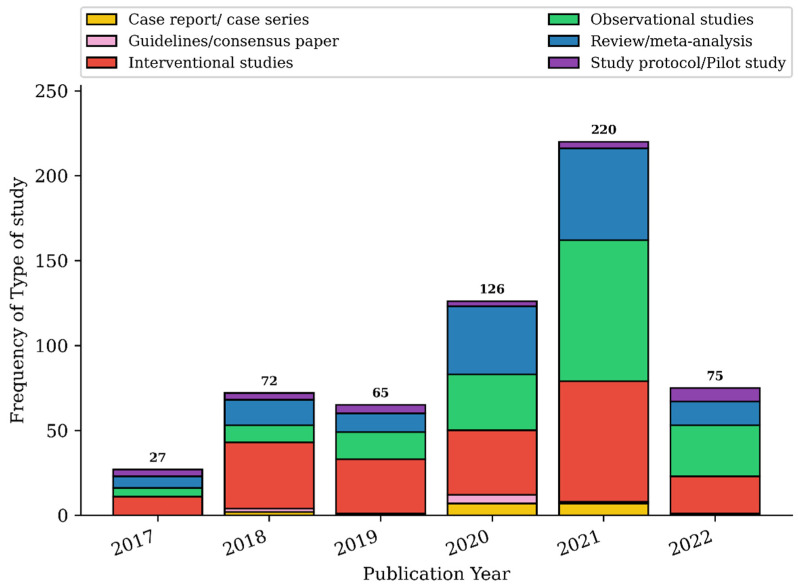
Frequency of type of studies according to publication year.

**Figure 4 ijerph-19-09787-f004:**
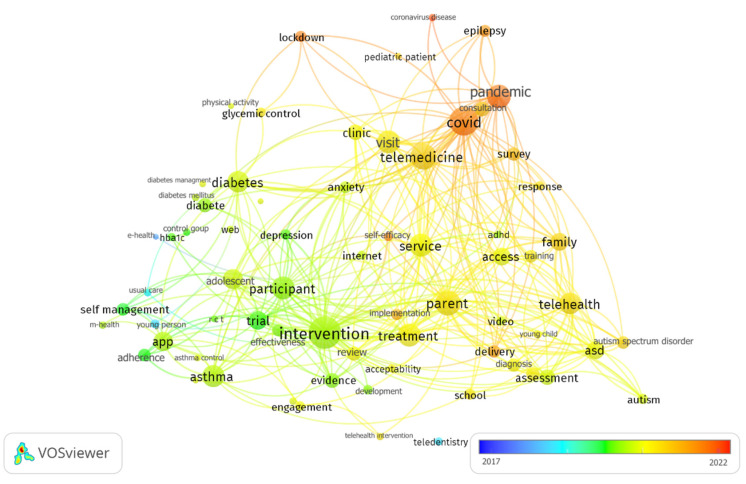
Bibliometric visualization of term co-occurrence map. Terms cited more than 200 times are shown according to the number of occurrences (circle size), year of highest occurrence (color), and co-occurrence between them (link).

**Table 1 ijerph-19-09787-t001:** Data comparison according to publication years.

Variable	Publication Years 2020–2022	Publication Years 2017–2019	*p*-Value
	(*n* = 421)	(*n* = 164)	
**Continent**			
Africa	3 (0.71%)	0 (0.00%)	0.001 *
Asia	46 (10.93%)	7 (4.26%)	
Europe	110 (26.13%)	42 (25.61%)	
Oceania	11 (2.61%)	17 (10.37%)	
North America	225 (53.44%)	88 (53.66%)	
South America	14 (3.33%)	5 (3.05%)	
More than one continent	12 (2.85%)	5 (3.05%)	
Total	421 (100.00%)	164 (100.00%)	
**Type of publication**			
Journal article	331 (78.62%)	132 (80.49%)	0.183
Book chapter	0 (0.00%)	1 (0.61%)	
Comment/editorial/letter	11 (2.62%)	7 (4.27%)	
Conference abstract	79 (18.76%)	24 (14.63%)	
Total	421 (100.00%)	164 (100.00%)	
**Topic**			
Asthma	77 (18.29%)	39 (23.78%)	<0.001 *
Brain disorders	44 (10.45%)	5 (3.05%)	
Dental caries	13 (3.09%)	12 (7.32%)	
Mental disorders	26 (6.18%)	18 (10.98%)	
Metabolic diseases	143 (33.97%)	57 (34.76%)	
Neurodevelopmental diseases or disorders	108 (25.65%)	25 (15.24%)	
Skin diseases	10 (2.37%)	8 (4.87%)	
Total	421 (100.00%)	164 (100.00%)	
**Type of study**			
Case report/case series	14 (3.33%)	3 (1.83%)	<0.001 *
Guidelines/consensus paper	7 (1.66%)	2 (1.22%)	
Interventional study	131 (31.12%)	82 (50.00%)	
Observational study	146 (34.68%)	31 (18.90%)	
Review/Meta-analysis	108 (25.65%)	33 (20.12%)	
Study protocol/pilot study	15 (3.56%)	13 (7.93%)	
Total	421 (100.00%)	164 (100.00%)	
**Clinical query**			
Prevention	9 (2.13%)	2 (1.22%)	0.109
Diagnosis	38 (9.03%)	17 (10.37%)	
Treatment	149 (35.39%)	73 (44.51%)	
Follow up	176 (41.81%)	50 (30.49%)	
Diagnosis and treatment/follow- up	49 (11.64%)	22 (13.41%)	
Total	421 (100.00%)	164 (100.00%)	

* *p*-Value <0.05 derived by 2-tailed Fisher’s exact test for categorical data.

**Table 2 ijerph-19-09787-t002:** Comparison of bibliometric variables according to publication years.

Variable	*n*	Mean	95% CI	SD	Range	*p*-Value
**Times cited count in Embase/WOS**						
2020–2022	421	5.54	4.23; 6.86	13.71	0.00–192.00	<0.001 *
2017–2019	164	14.71	12.07; 17.36	17.16	0.00–79.00	
**2020 Impact Factor**						
2020–2022	359	4.58	4.28; 4.87	2.82	0.06–21.60	0.709
2017–2019	135	5.46	4.67; 6.24	4.62	0.27–32.07	

## Data Availability

All data supporting the results are reported in the manuscript or in the Appendix A.

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
