# Peer review of "COVID-19 Pandemic and Remote Consultations in Children: A Bibliometric Analysis"

_ijerph, 2022, doi:10.3390/ijerph19169787_

Round 1
Reviewer 1 Report
Introduction
- The management of the COVID-19 pandemic has relied on cautious contact tracing, quarantine, and sterilization protocols while we await a vaccine to be made widely available. Telemedicine or mobile health (mHealth) is well-positioned during this time to reduce potential disease spread and prevent overloading of the healthcare system through at-home COVID-19 screening, diagnosis, and monitoring. With the rise of mass-fabricated electronics for wearable and portable sensors, emerging telemedicine tools have been developed to address shortcomings in COVID-19 diagnostics, monitoring, and management, to better control the COVID-19 pandemic. please discuss and cite doi:10.1021/acsnano.0c08494. and doi:10.23750/abm.v92i1.11281.
- add the joanna briggs tool to assess bias
Author Response
- The management of the COVID-19 pandemic has relied on cautious contact tracing, quarantine, and sterilization protocols while we await a vaccine to be made widely available. Telemedicine or mobile health (mHealth) is well-positioned during this time to reduce potential disease spread and prevent overloading of the healthcare system through at-home COVID-19 screening, diagnosis, and monitoring. With the rise of mass-fabricated electronics for wearable and portable sensors, emerging telemedicine tools have been developed to address shortcomings in COVID-19 diagnostics, monitoring, and management, to better control the COVID-19 pandemic. please discuss and cite doi:10.1021/acsnano.0c08494. and doi:10.23750/abm.v92i1.11281.
Thank you for the suggestion; we agree that monitoring technologies for COVID-19 should have been more emphasized, as tracking systems are still a major issue in this post-pandemic period. We have added a discussion on this from line 62.
The first article by Lukas et al. was already cited in line 48, but we have cited it again in this new paragraph.
The article by Spinato et al. was added, discussed and cited in line 57.
- add the joanna briggs tool to assess bias
We believe you suggested using the JBI tool because we defined our study as qualitative. After your review, we discussed in depth what our objective was and what we wanted to add to the current literature on COVID-19. Our conclusion was that our article does not present a qualitative, but a quantitative analysis of the literature and we therefore corrected the statement in line 111. We actually followed the methodological framework for the scoping review by Arksey and O'Malley. We have added a paragraph from line 95 about how the scoping method, being relatively new, follows different review paths from those used in systematic reviews. Indeed, our article tries to be very practical by providing a snapshot of the literature on NCDs in paediatric populations and telemedicine. This article shows an overall state of the art scientific contribution in terms of which paediatric NCDs have already been studied and are already supported by remote technologies, highlighting the benefits for both patients and clinicians, such as in the case of diabetes or asthma. The study also highlights the need for further studies that support the use of telemedicine in as yet under-researched areas such as oral health and caries in particular. We realize that qualitative studies allow for higher scientific standards, but the purpose with which we have carried out this research is to assess how the pandemic has contributed to changing certain medical contexts such as that of non-communicable diseases that are widespread in a large part of the global paediatric population.
English language was revised
Reviewer 2 Report
Thank you for submitting the paper "Covid-19 outbreak and remote consultations: how pandemic boosted telemedicine literature about NCDs in children. A bibliometric analysis".
This is an interesting and well organized article, just a few observations.
-7 months have passed since the bibliographic search, it must be updated.
-Why papers written in any languages with an abstract in English? You only read the abstract?
-Exclusion criteria: “Studies that considered both paediatric and adult population”. In inclusion criteria you should specify which should be considered.
-Search strategy should be on paper not in additional files.
Author Response
Thank you for submitting the paper "Covid-19 outbreak and remote consultations: how pandemic boosted telemedicine literature about NCDs in children. A bibliometric analysis".
This is an interesting and well organized article, just a few observations.
- 7 months have passed since the bibliographic search, it must be updated.
Thank you for your comment. We decided to consider 2 years before and 2 years from the onset of the pandemic in order to have 2 similar time periods large enough to study the phenomenon at hand, i.e. the change in the use of telemedicine for the most common paediatric NCDs before and after the onset of the pandemic. Therefore, an update did not seem essential for this purpose.
- Why papers written in any languages with an abstract in English? You only read the abstract?
Yes, because we only wanted to know the bibliometric data and the main topics studied in each article. We used Arksey and O'Malley's methodological framework for scoping review which clearly states that:
"The aim of a scoping review is to examine the extent, range and nature of the research activity: this type of rapid review may not describe the research findings in detail, but it is a useful way of mapping fields of study where it is difficult to visualise the range of material that might be available." Indeed, our article tries to be very practical by providing a snapshot of the literature on NCDs in paediatric populations and telemedicine. This article shows an overall state of the art scientific contribution in terms of which paediatric NCDs have already been studied and are already supported by remote technologies, highlighting the benefits for both patients and clinicians, such as in the case of diabetes or asthma. The study also highlights the need for further studies that support the use of telemedicine in as yet under-researched areas such as oral health and caries in particular. We realise that qualitative studies allow for higher scientific standards, but the purpose with which we have carried out this research is to assess how the pandemic has contributed to changing certain medical contexts such as that of non-communicable diseases that are widespread in a large part of the global paediatric population.
- Exclusion criteria: “Studies that considered both paediatric and adult population”. In inclusion criteria you should specify which should be considered.
We agree; it was added at line 121
- Search strategy should be on paper not in additional files.
The search strategy was added from line 146
Reviewer 3 Report
Thank you for submitting your manuscript. This is an interesting study with important topic especially it was conducted during COVID 19 pandemic. However, The manuscript will benefit from general editing and revising, there are many issues with writing and some information need to be clarified. See below my suggestions:
Suggest to make the title shorter, you should not include abbreviation (NCD) in title and COVID 19 is a pandemic not an outbreak, may be revise to:
Covid-19 pandemic and remote consultations in children: A bibliometric analysis.
Revise outbreak to pandemic throughout the manuscript
Line 42, the outbreak starts in 2019
Line 55, moved (NCDs) before “in..” and after disease
Line 78 delete PICO
Line 90 my understanding you included studies that have an abstract in English, so if the abstract was in English and the manuscript was in other language how did you understand the findings? did you use a translation service (this should be included in the limitation)
What is your inclusions criteria in relation to population?
Line 107 what is NC, AB
Line 146 did not
Table 1 one of your variable is continent, is multicenter a continent?
Fig 4 needs to be revised it is unclear, you should also include description to explain the figure
There are many limitation in this study, include a limitation section in the end of the discussion.
Author Response
Thank you for submitting your manuscript. This is an interesting study with important topic especially it was conducted during COVID 19 pandemic. However, the manuscript will benefit from general editing and revising, there are many issues with writing and some information need to be clarified. See below my suggestions:
Suggest to make the title shorter, you should not include abbreviation (NCD) in title and COVID 19 is a pandemic not an outbreak, may be revise to:
- Covid-19 pandemic and remote consultations in children: A bibliometric analysis.
We agree with the suggestion. The title was changed.
- Revise outbreak to pandemic throughout the manuscript
It was revised.
- Line 42, the outbreak starts in 2019
It was updated at line 48.
- Line 55, moved (NCDs) before “in..” and after disease
It was modified.
- Line 78 delete PICO
Done.
- Line 90 my understanding you included studies that have an abstract in English, so if the abstract was in English and the manuscript was in other language how did you understand the findings? did you use a translation service (this should be included in the limitation)
This is a bibliometric analysis and what we only wanted to know is the bibliometric data and the main topics studied in each article. For this reason, the abstracts were used.
We used Arksey and O'Malley's methodological framework for scoping review which clearly states that:
"The aim of a scoping review is to examine the extent, range and nature of the research activity: this type of rapid review may not describe the research findings in detail, but it is a useful way of mapping fields of study where it is difficult to visualise the range of material that might be available." Indeed, our article tries to be very practical by providing a snapshot of the literature on NCDs in paediatric populations and telemedicine. This article shows an overall state of the art scientific contribution in terms of which paediatric NCDs have already been studied and are already supported by remote technologies, highlighting the benefits for both patients and clinicians, such as in the case of diabetes or asthma. The study also highlights the need for further studies that support the use of telemedicine in as yet under-researched areas such as oral health and caries in particular. We realise that qualitative studies allow for higher scientific standards, but the purpose with which we have carried out this research is to assess how the pandemic has contributed to changing certain medical contexts such as that of non-communicable diseases that are widespread in a large part of the global paediatric population.
- What is your inclusions criteria in relation to population?
Inclusion criteria were modified, including target population
- Line 107 what is NC, AB
They are the acronyms of the names of two of the authors, we changed the text to be clearer.
- Line 146 did not
It was modified.
- Table 1 one of your variable is continent, is multicenter a continent?
Sorry for this. Table 1 was correct
English was revised
Round 2
Reviewer 2 Report
8 months have passed since the bibliographic search, it MUST be updated.
Author Response
Dear Reviewer, we updated the search as requested. We decided, in order to better compare the periods before/after covid-19 pandemic, to search from 1st July 2017 to 1 August 2022. This update allowed us to add 101 records in the included section. All the tables and figures were updated.
Reviewer 3 Report
The author makes the required edits and the manuscripts is in better shape.
Author Response
We thank the reviewer for his help in improving the paper